# Correlated states controlled by a tunable van Hove singularity in moiré WSe$_2$ bilayers

Patrick Knüppel [1,5], Jiacheng Zhu[1,5], Yiyu Xia[1], Zhengchao Xia [1], Zhongdong Han[1], Yihang Zeng [1], Kenji Watanabe [2], Takashi Taniguchi [2], Jie Shan [1,3,4] ✉ & Kin Fai Mak [1,3,4] ✉

Twisted transition metal dichalcogenide (TMD) bilayers have enabled the discovery of superconductivity, ferromagnetism, correlated insulators, and a series of new topological phases of matter. However, the connection between these electronic phases of matter and the underlying band structure singularities has remained largely unexplored. Here, combining magnetic circular dichroism and exciton sensing measurements, we investigate the influence of a van Hove singularity (vHS) on the correlated phases in bilayer WSe$_2$ with twist angle between 2 and 3 degrees. By tuning the vHS across the Fermi level using electric and magnetic fields, we observe Stoner ferromagnetism below moiré lattice filling one and Chern insulators at filling one. The experimental observations are supported by the continuum model band structure calculations. Our results highlight the prospect of engineering electronic phases of matter in moiré materials by tunable van Hove singularities.

In two dimensions, a divergence in the density of states (DOS), also known as a van Hove singularity (vHS), arises from a saddle point in the electronic band structure. The presence of a vHS near the Fermi level significantly enhances the electron-electron interactions which often lead to electronic instabilities and new phases of matter[1,2]. However, the Fermi level in conventional bulk materials cannot be easily tuned to cross the vHS. The emergence of moiré materials[3–7], which support the highly tunable Fermi levels and electronic band structures, has provided a platform to engineer electronic phases of matter by bringing together the vHS and the Fermi level.

Among the moiré materials, twisted homo-bilayers of TMDs, such as MoTe$_2$ and WSe$_2$, have recently attracted significant attention because they possess flat moiré bands with finite valley-contrasting Chern numbers[8–15]. A suite of correlated and topological states, including the integer[16] and fractional Chern insulators[17–20], integer[21] and fractional quantum spin Hall insulators[22], superconductivity[23,24], ferromagnetism[25] and metal-insulator transitions[26–29], has been reported. The electronic band structure calculations show a saddle point in the topmost moiré valence band located at the $m$-point of the moiré Brillouin zone. The calculations also show that an electric field

perpendicular to the sample plane, which controls the interlayer potential difference[8,9,30], can widely tune the electronic band structure including the location of the vHS[26,28–32]. It has been suggested that proximity of the vHS to the Fermi level can affect the stability of the correlated insulators at integer moiré lattice fillings[10,30,31,33]. However, the general effect of the van Hove singularities on symmetry-breaking ground states at generic fillings has remained elusive.

Here we investigate hole-doped twisted WSe$_2$ (tWSe$_2$) bilayers with twist angle between 2 and 3 degrees. We demonstrate a ferromagnetic metal phase below filling one and a Chern insulator at filling one by tuning the vHS across the Fermi level using the electric and magnetic fields. These states are identified by combining reflective magnetic circular dichroism (MCD)[34] and exciton sensing measurements[35]. Specifically, the former probes the valley polarization, which is connected to the spin polarization, because of spin-momentum locking—a property inherited from the transition metal dichalcogenide (TMD) monolayers[36]. The latter probes the sample's electronic incompressibility through the sensitivity of the sensor excitons to their dielectric environment; it also determines the Chern number of the insulating states through their dispersion with an

[1]Laboratory of Atomic and Solid-State Physics and School of Applied and Engineering Physics, Cornell University, Ithaca, NY, USA. [2]National Institute for Materials Science, Tsukuba, Japan. [3]Kavli Institute at Cornell for Nanoscale Science, Ithaca, NY, USA. [4]Max Planck Institute for the Structure and Dynamics of Matter, Hamburg, Germany. [5]These authors contributed equally: Patrick Knüppel, Jiacheng Zhu. ✉e-mail: jie.shan@cornell.edu; kinfai.mak@cornell.edu

externally applied magnetic field, as described by the Streda formula. Our results elucidate the impact of the vHS on symmetry-breaking ground states at generic doping densities in moiré materials and provide insight into how to design robust magnetism in correlated materials in general.

## Results and discussion

### Phase diagram of tWSe₂

Figure 1a illustrates the dual-gated device structure of tWSe₂ employed in this study. The top and bottom gate voltages ($V_{tg}$ and $V_{bg}$) independently control the moiré lattice filling factor ($\nu$) and the electric field ($E$) perpendicular to the sample plane. (Filling factor $\nu = 1$ is defined as one hole per moiré unit cell, which corresponds to a half-filled moiré valence band.) A WS₂ monolayer, separated from the sample by a thin hexagonal boron nitride (hBN) spacer (about 1 nm thick), is used as the exciton sensor. We first focus on a 2.7° tWSe₂ sample and demonstrate the effect of twist angle at the end. Unless otherwise specified, all results are obtained at sample temperature $T = 1.6$ K. See Methods for details on the device fabrication, band structure calculations, and optical measurements.

Twisted WSe₂ bilayers form a honeycomb or triangular moiré lattice depending on the twist angle[37]. The schematic in Fig. 1a illustrates a honeycomb moiré lattice with two sublattices centered at the MX and XM (M = W; X = Se) stacking sites. Figure 1b is the electronic band structure (left panel) and the energy-dependent DOS (right panel) for $E = 0$ calculated using the continuum model described in Refs. 8,9. Only the first two moiré valence bands of the K-valley states are illustrated. They carry Chern number $C = +1$. The corresponding moiré bands of the K′-valley states are a time-reversal copy and carry Chern number $C = -1$. The DOS shows a vHS that is aligned with the saddle point in the first moiré band, and a second vHS, with the saddle point in the second moiré band. The Fermi level crosses the first vHS at hole filling $\nu \approx 0.7$. Supplementary Fig. 1 shows the band structure

under large electric fields, where the two layers are decoupled and the bands are more dispersive.

Figure 1c, d are the phase diagrams of tWSe₂ as a function of $\nu$ and $E$. We determine $\nu$ and $E$ using the applied gate voltages and gate capacitances, which are calibrated using the observed quantum oscillations under high magnetic fields perpendicular to the sample plane (Methods and Supplementary Fig. 2). Figure 1c shows the spectrally averaged reflection contrast (RC) of the intralayer exciton resonances of the sample. (Representative raw spectra are included in Supplementary Fig. 3). We use it to identify the layer-hybridized region for small electric fields and the layer-polarized region for large electric fields. Because the intralayer exciton resonance of a layer is sensitive to doping in the layer, it shows the strongest $E$ dependence at the boundary between the two regions (dashed lines), across which one of the layers becomes undoped. The electric field at the boundary increases with $\nu$ because the electrostatics requires a larger electric field to fully polarize a higher charge density to one of the layers. The phase diagram is symmetric about $E = 0$ after removing a small build-in field of $E_0 \approx 13$ mV/nm (likely from the device structure asymmetry). These results are fully consistent with the reported phase diagram of tWSe₂ (Refs. 23,24) and tMoTe₂ (Refs. 8,9,17,18).

Figure 1d shows the spectrally averaged RC of the sensor 2 s exciton. Representative raw spectra as a function of filling factor are shown in Supplementary Fig. 4 (see Methods for spectral analysis). Throughout the measurement, the WS₂ sensor, which has a type-II band alignment with WSe₂ (Ref. 38), has been kept charge neutral. An incompressible state in the sample manifests a blue shift and an enhanced spectral weight of the sensor 2 s exciton due to the reduced dielectric screening[35,39]. We identify several correlated insulators at commensurate fillings, $\nu = 1$, 1/3, 1/4, and 1/6, in the layer-hybridized region. These states become compressible in the layer-polarized region, where the moiré bands are more dispersive and the correlation effects are weaker. The phase diagram appears

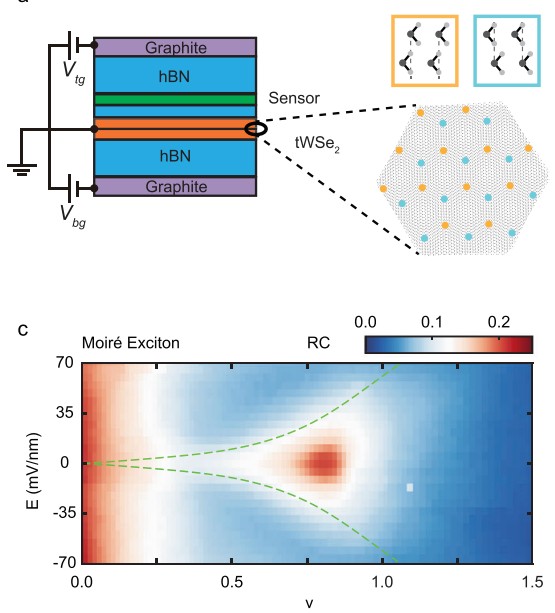

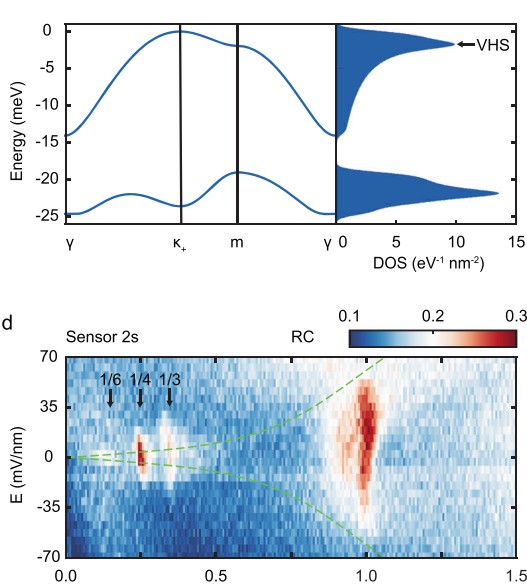

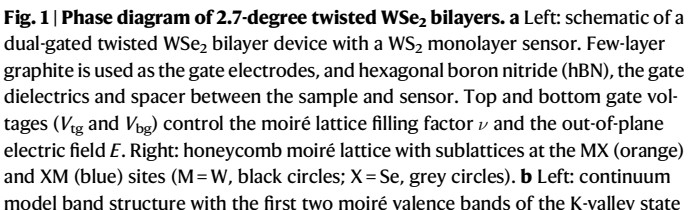

**Fig. 1 | Phase diagram of 2.7-degree twisted WSe₂ bilayers. a** Left: schematic of a dual-gated twisted WSe₂ bilayer device with a WS₂ monolayer sensor. Few-layer graphite is used as the gate electrodes, and hexagonal boron nitride (hBN), the gate dielectrics and spacer between the sample and sensor. Top and bottom gate voltages ($V_{tg}$ and $V_{bg}$) control the moiré lattice filling factor $\nu$ and the out-of-plane electric field $E$. Right: honeycomb moiré lattice with sublattices at the MX (orange) and XM (blue) sites (M = W, black circles; X = Se, grey circles). **b** Left: continuum model band structure with the first two moiré valence bands of the K-valley state

along the directions of $\gamma - \kappa - m - \gamma$ in the moiré Brillouin zone ($E = 0$). Right: density of states (DOS) showing van Hove singularities (vHS). **c, d** Spectrally integrated reflection contrast (RC) of the sample intralayer exciton (**c**) and the sensor 2 s exciton (**d**) as a function of filling factor $\nu$ and electric field $E$ at 1.6 K. The dashed lines extracted from (**c**) (see main text) separate the layer-hybridized region for low fields from the layer-polarized region for high fields. Black arrows mark the correlated insulators at fractional filling factors.

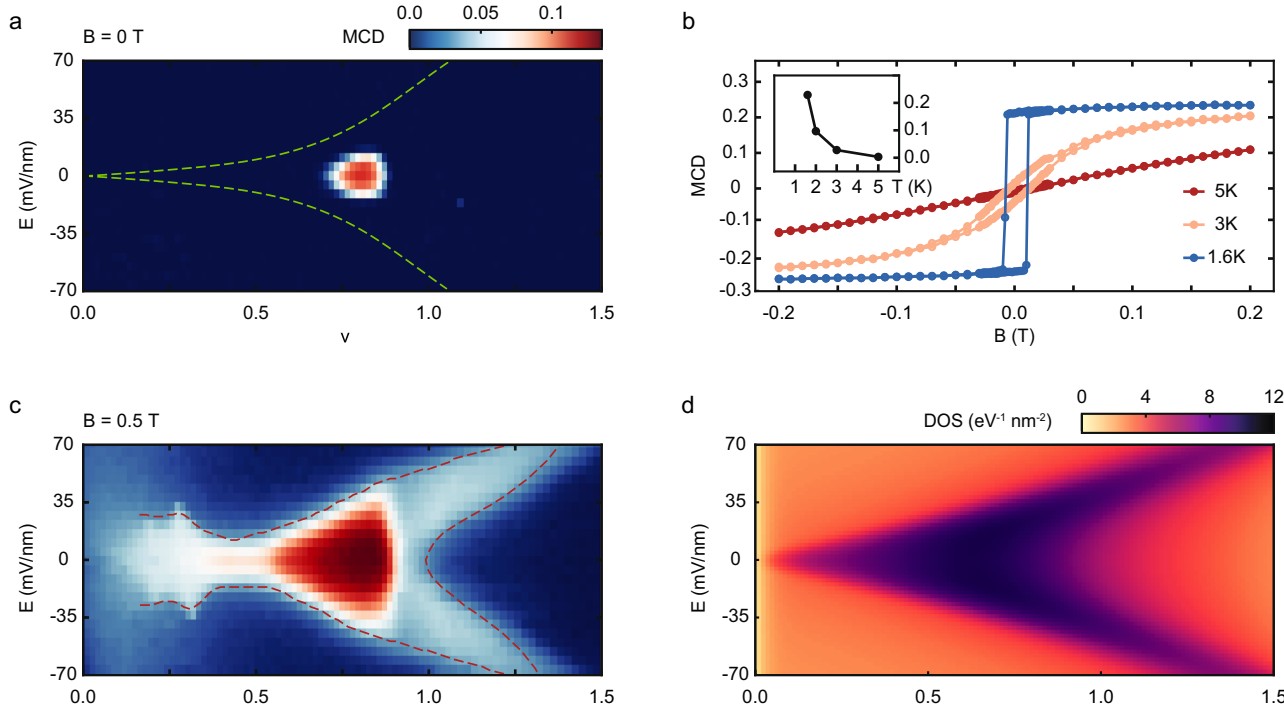

**Fig. 2 | Stoner ferromagnetism. a** Spontaneous magnetic circular dichroism (MCD) as a function of moiré lattice filling factor $\nu$ and out-of-plane electric field $E$ at 1.6 K. The dashed lines (from Fig. 1c) separate the layer-hybridized and layer-polarized regions. **b** Magnetic-field dependence of MCD of the hot spot in (**a**) at representative temperatures. Clear magnetic hysteresis is observed at low temperatures. Inset: temperature dependence of the spontaneous MCD. **c** Same as (**a**) under magnetic field $B = 0.5$ T. The dashed lines are a guide to the eye of the boundary of the region with enhanced magnetic susceptibility. **d** Calculated electronic density of states (DOS) as a function of $\nu$ and $E$ showing the evolution of the van Hove singularity (vHS) with $E$. All results are for 2.7° twisted WSe$_2$.

asymmetric about $E = 0$ because the sensor is placed above the top WSe$_2$ layer.

## Stoner ferromagnetism

We examine the magnetic properties of tWSe$_2$ by performing the MCD measurements (see Supplementary Fig. 4 for MCD spectra around the intralayer exciton resonances and Methods for extracting the spectrally averaged MCD, which we refer to as MCD below). Figure 2a shows MCD as a function of $\nu$ and $E$ in the absence of magnetic fields. We observe a hot spot of spontaneous MCD around $\nu = 0.8$ and $E = 0$. It corresponds to a compressible region of the phase diagram (Fig. 1d). The MCD exhibits a clear magnetic hysteresis with a coercive field of about 10 mT (Fig. 2b). As temperature increases, the spontaneous MCD and the magnetic hysteresis gradually weaken and vanish above about 3 K (inset of Fig. 2b). These results support that the MCD hot spot corresponds to a ferromagnetic metal.

Figure 2c is MCD under a small out-of-plane magnetic field of $B = 0.5$ T. The result at higher magnetic fields is included in Supplementary Fig. 5. Except in the ferromagnetic region, the MCD increases linearly with magnetic field for small fields, and is proportional to the magnetic susceptibility[22]. The susceptibility is substantially enhanced in the layer-hybridized region below filling one. It is also higher in regions near the correlated insulators at $\nu = 1/3$, 1/4, and 1/6. Above filling one, the region with enhanced susceptibility disperses with electric field, exhibiting an arrowhead-like feature. The dashed lines, at which the MCD drops to 0.03, provide a guide to the eye of the boundary of this region.

To gain insight into the magnetic properties of tWSe$_2$, we perform band structure calculations under varying electric fields. Figure 2d illustrates the electronic DOS as a function of $\nu$ and $E$ in the experimentally relevant region of the phase diagram. Line cuts at

representative electric fields are shown in Supplementary Fig. 1. High DOS is observed in the layer-hybridized region below filling one, where the moiré bands are relatively flat. The vHS is located near $\nu = 0.7$ for $E = 0$; it continuously shifts towards higher filling factors with reduced DOS as electric field increases. The evolution of the vHS with electric field has been verified by transport measurements[23,26,28] (Supplementary Fig. 6) although the precise location of the vHS in ($\nu$, $E$) is dependent on the twist angle, the choice of the continuum model parameters, and possibly also the interaction effects which are not accounted for in the single-particle continuum model.

The measured magnetic susceptibility is well correlated with the calculated electronic DOS except the regions near the correlated insulators and is substantially enhanced at the vHS. This is expected because the magnetic susceptibility of a Landau Fermi liquid is proportional to the electronic DOS[40]. (In the correlated insulators, the band picture breaks down, and the susceptibility arises from the local magnetic moments, that is, the spins of the localized holes). Remarkably, ferromagnetic order is stabilized near the vHS around $E = 0$ where the DOS is the highest. The magnetic phase space is much smaller than that of the enhanced DOS or susceptibility. This supports the Stoner mechanism: the ferromagnetic metal is driven by strong Coulomb repulsion and enhanced by high DOS. The Stoner criterion[1], $UD_F > 1$, expressed in terms of the strength of Coulomb repulsion $U$ and the single-particle DOS at the Fermi level $D_F$, provides a qualitative threshold for magnetism. This picture is further supported by the observation of a second ferromagnetic metal phase near the second vHS in tWSe$_2$ around $E = 0$ (Supplementary Fig. 7).

## Chern insulators

Next, we examine the effect of an out-of-plane magnetic field on the correlated insulators at $\nu = 1$. We focus on the case of $E = 0$. Figure 3a

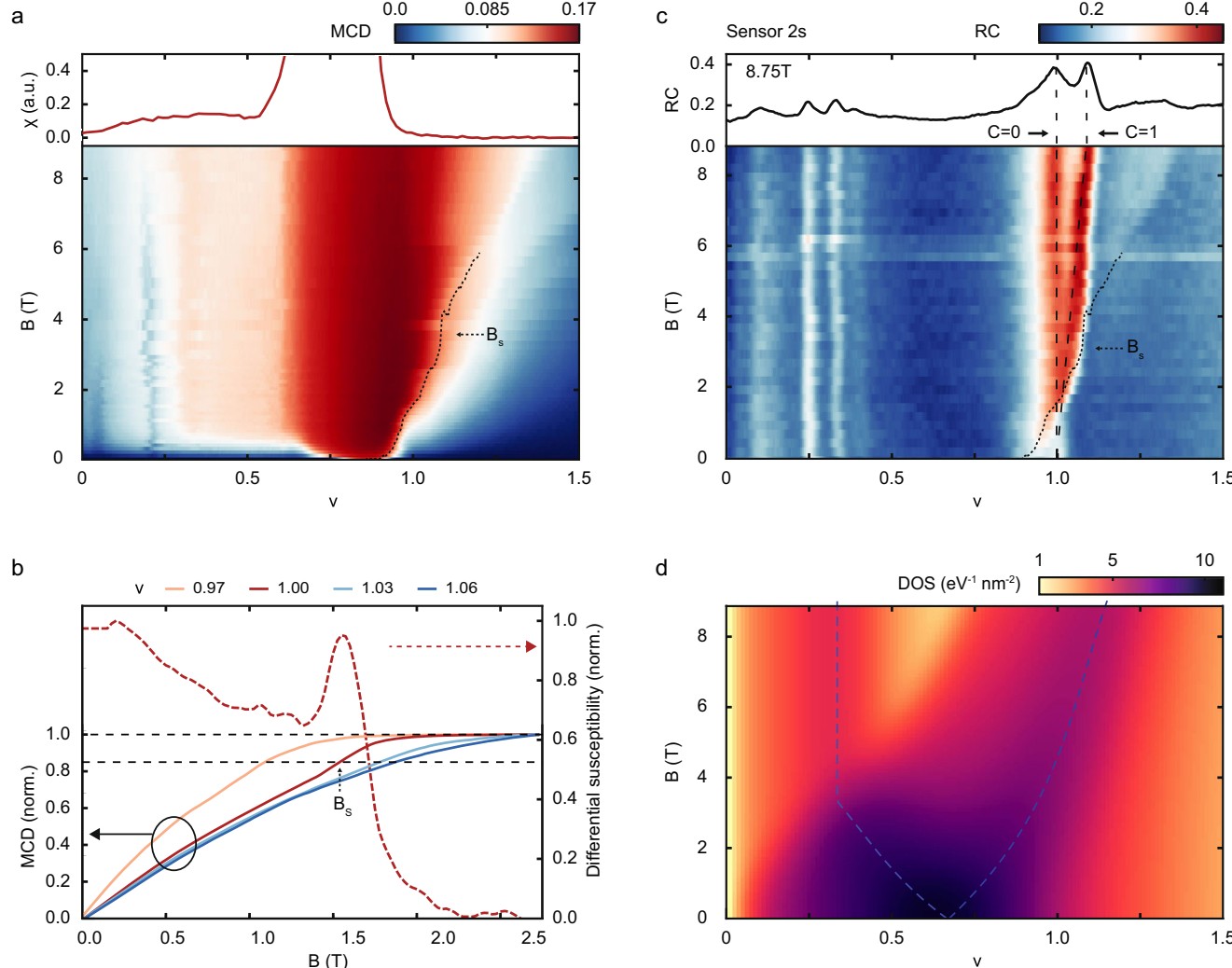

**Fig. 3 | Magnetic-field tuned Chern insulator. a** Magnetic circular dichroism (MCD) as a function of moiré lattice filling factor $\nu$ and out-of-plane magnetic field $B$ under zero electric field. Top: filling factor dependence of magnetic susceptibility $\chi$ extracted from the small-field MCD. **b** Magnetic-field dependence of the normalized MCD for representative filling factors around one (solid lines, left axis) and the derivative of MCD for $\nu = 1$ (dashed red line, right axis). MCD reaches 85% of the fully saturated value (dashed lines) at the saturation field $B_s$. **c** Spectrally integrated optical reflection contrast (RC) of the sensor 2 s exciton as a function of $\nu$ and $B$. The dashed lines show the expected dispersion for states with Chern number $C = 0$ and 1 near $\nu = 1$. Top: line cut at the highest field of $B = 8.75$ T. The dotted line in (**a, c**) denotes the saturation field $B_s$. **d** Calculated electronic density of states (DOS) as a function of $\nu$ and $B$ showing Zeeman splitting of the van Hove singularity (vHS, dashed lines). All measurements were performed at 1.6 K. All results are for 2.7° twisted WSe$_2$.

displays MCD as a function of $\nu$ and $B$ (lower panel) and magnetic susceptibility as a function of $\nu$ (upper panel). Away from the ferromagnetic metal around $\nu = 0.8$, MCD increases with magnetic field till full spin/valley polarization is reached. (The ferromagnetic metal shows spontaneous MCD and diverging susceptibility.) Fig. 3b illustrates the field dependence of normalized MCD by the saturation value for several filling factors around one. We quantify the saturation field $B_s$ by using the value at which the normalized MCD reaches 0.85. The saturation field increases rapidly with filling factor, from zero near $\nu = 0.8$ to several tesla above $\nu = 1$. At $B_s$, the Zeeman energy is sufficient to overcome the exchange energy for full spin/valley polarization. The latter has been independently estimated through the Curie-Weiss analysis of the temperature dependence of the magnetic susceptibility (Methods). In addition, there is a weak kink in the field dependence of MCD before saturation for fillings around one. It is more clearly seen in the derivative of MCD (dashed red line) as a local minimum followed by a maximum near $B_s$. This suggests a metamagnetic transition and the transition field is close to $B_s$.

Figure 3c displays the evolution of the correlated insulators at $\nu = 1$ with magnetic field. The lower panel shows the sensor 2 s exciton response as a function of $\nu$ and $B$; the upper panel is a line cut at the highest applied field of $B = 8.75$ T; the dotted line denotes the saturation field. For small fields, the insulator at $\nu = 1$ does not disperse with magnetic field. Hence, it has Chern number $C = 0$ based on the Streda formula and is topologically trivial. For fields above $B_s$, the state turns into two insulators with $C = 0$ and 1. A potential candidate for the low-field state is a valley-coherent insulator[30], whereas the emergence of the Chern insulator with $C = 1$ at high fields is compatible with the spin/valley-contrasting Chern bands in 2.7° tWSe$_2$. The coexistence of two types of insulators after magnetic saturation suggests that these competing states are close in their ground state energies. This is distinct from tMoTe$_2$, where only the Chern insulator is observed regardless of the field (Refs. [17,18]). (The insulating states at fractional filling factors have $C = 0$ and are likely generalized Wigner crystals[35,41].)

We perform the continuum model band structure calculations under varying magnetic fields to elucidate the effect of the vHS on the

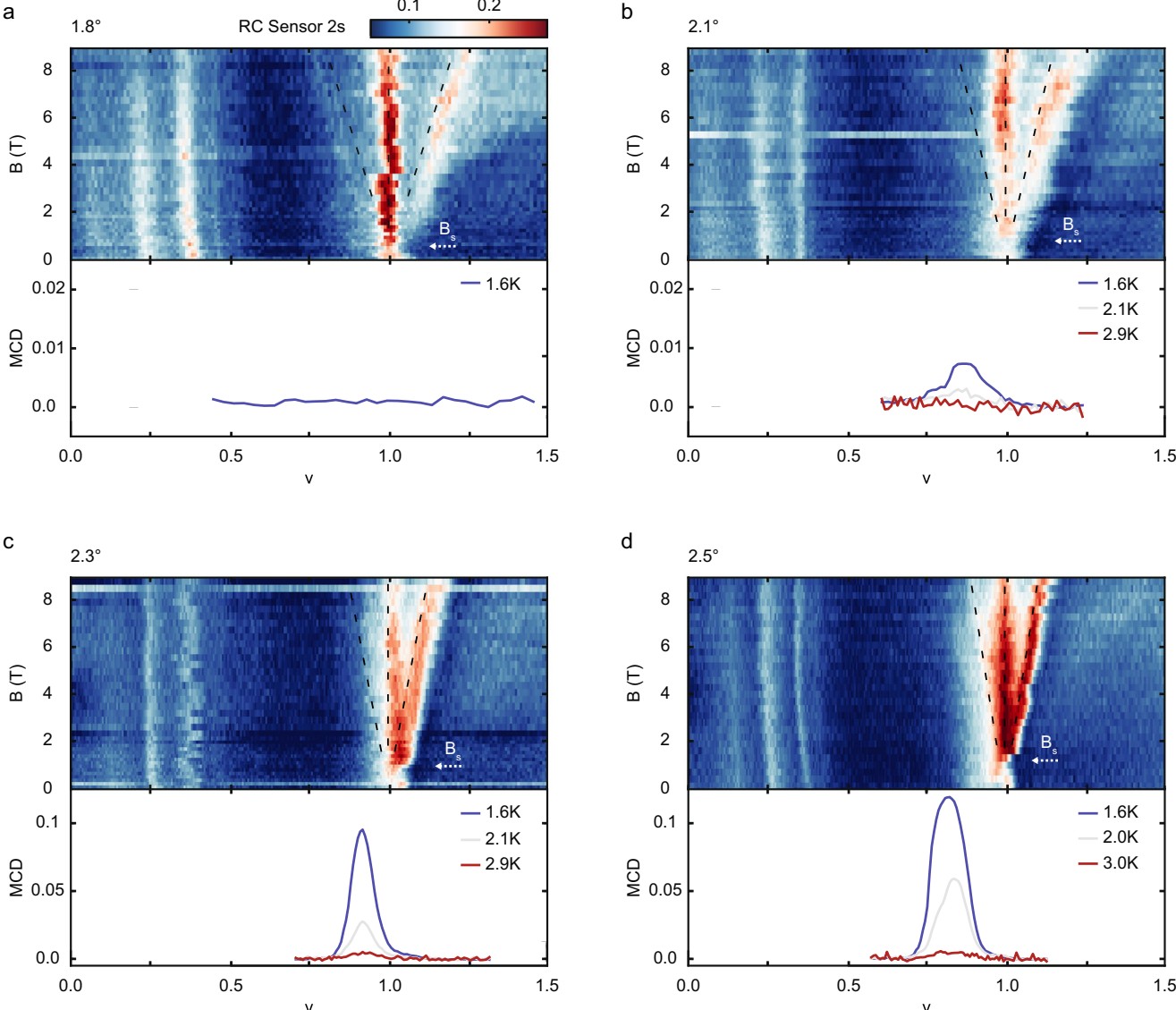

**Fig. 4 | Twist angle dependence. a–d** Top: Spectrally integrated optical reflection contrast (RC) of the sensor 2 s exciton as a function of moiré lattice filling factor $\nu$ and out-of-plane magnetic field $B$ under zero electric field at 1.6 K. The dashed lines show the expected dispersion for states with Chern number $C = -1$, 0 and 1 near $\nu = 1$. The arrows denote the saturation field and emergence of the Chern insulator(s). Bottom: filling dependence of the spontaneous magnetic circular dichroism (MCD) at representative temperatures. The twist angle in WSe$_2$ bilayers is 1.8 (**a**), 2.1 (**b**), 2.3 (**c**) and 2.5 degrees (**d**). The saturation field $B_s$ (denoted by white arrows) decreases with twist angle. In 1.8° twisted WSe$_2$, spontaneous MCD is not observed down to 1.6 K and a weak $C = -1$ Chern insulator coexists with a $C = 1$ Chern insulator.

correlated insulators at $\nu = 1$. For simplicity, we only consider the Zeeman effect. Figure 3d illustrates the calculated electronic DOS as a function of $\nu$ and $B$. The vHS with large DOS (dashed lines) splits into two under $B$. These features have also been identified in transport studies[1,29]. The vHS that disperses to higher filling factors crosses $\nu = 1$ at about $B \approx 4$ T. The qualitative agreement between this field and the saturation field suggests a scenario in which the vHS tuned to the Fermi level (for $\nu = 1$) by the magnetic field induces the transition from the spin/valley unpolarized to polarized states and the emergence of the Chern insulator.

**Twist angle dependence**

Finally, we demonstrate that the observed effect of a vHS near the Fermi level on the correlated states is general for tWSe$_2$ of small twist angles. Figure 4a–d display the evolution of the correlated insulators with magnetic field (upper panel) and the filling dependence of spontaneous MCD at representative temperatures (lower panel) in tWSe$_2$ with twist angle 1.8 (a), 2.1 (b), 2.3 (c), and 2.5 degrees (d). The

black dashed lines denote the expected dispersion for states with Chern number 0 and ±1 near $\nu = 1$. The results show that Stoner magnetism is stabilized in all but the lowest twist angle sample (1.8°). Chern insulators are absent at zero magnetic field; they emerge above the saturation field for all twist angles. In general, decreasing the twist angle lowers the saturation field (denoted by arrows for $\nu = 1$) obtained from the MCD measurements (Supplementary Fig. 8). In addition, in 1.8° tWSe$_2$, a weak $C = -1$ Chern insulator coexists with the $C = +1$ Chern insulator, whereas only the $C = +1$ Chern insulator is stable for samples with larger twist angles.

In summary, combining observations from the exciton sensing and magnetization measurements with the continuum model band structure calculations, we develop a general understanding of the electronic phase diagram for tWSe$_2$ and connection to the vHS in the band structure. Our work bridges the gap[37,42] between the previously explored regime of small twist angles (with strong correlation effects)[16] and regime of large twist angles (with weak correlation effects)[26,28,29]. It also provides the basis for vHS engineering of the correlated states in

moiré material, such as superconductivity and exciton condensation involving flat Chern bands[43].

## Methods

### Device fabrication

Twisted $WSe_2$ moiré devices were assembled using the layer-by-layer dry transfer method[44]. Thin flakes of hBN and graphite, monolayer $WSe_2$, and monolayer $WS_2$ were exfoliated onto Si/SiO$_2$ substrates. Optical RC was used to identify the appropriate flake shape and thickness. A thin film of polycarbonate on polydimethylsiloxane was employed as a stamp to pick up the flakes following the sequence shown in Fig. 1a. The complete stack was released at 180 °C onto a Si/SiO$_2$ substrate with prepatterned platinum (Pt) gate electrodes. To create the moiré superlattice, a flake of monolayer $WSe_2$ was cut into two parts using an atomic force microscope tip and stacked with a small relative twist angle of $\theta$.

### Optical characterizations (RC and MCD)

Optical measurements were performed in a closed-cycle cryostat (Attocube, Attodry 2100) with magnetic fields up to 9T and temperatures down to 1.6 K. Either a halogen lamp (for 2 s sensing and MCD) or a light emitting diode (LED, for MCD) was used as the light source. The input light was spatially filtered by a single-mode fiber and sent into the cryostat as a collimated beam. A low-temperature microscope objective (Attocube, numerical aperture 0.8) was used to focus the light onto the sample. The light intensity on the sample was kept below 50 nW/μm$^2$ to minimize its effects on the electronic states; negligible changes in the magnetization were observed by further reducing the incident power by an order of magnitude. The reflected light was collected by the same objective and analyzed by a spectrometer equipped with a liquid-nitrogen-cooled charge coupled device array to obtain spectrum $R$. The RC spectrum is defined as $((R - R_0)/R_0$, where the reference spectrum $R_0$ was taken for sample at a high doping density with quenched excitonic resonances.

The reflective MCD was used to study the magnetic properties of the samples. A combination of a linear polarizer and a quarter-wave plate was used to generate a right and left circularly polarized light ($\sigma^-$ and $\sigma^+$) on the sample. The MCD spectrum is defined as $(R^- - R^+)/(R^- + R^+)$, where $R^-$ and $R^+$ are the reflection spectra for the $\sigma^-$ and $\sigma^+$ incident light, respectively.

The RC spectrum of tWSe$_2$ depends sensitively on the doping density, displacement field, and magnetic field. To account for these changes, we chose to average the reflectance (for combined $\sigma^-$ and $\sigma^+$ channels) over a range of wavelength (725–745 nm, or equivalently, 1.66–1.71 eV) that focuses on the 1 s exciton resonance of tWSe$_2$. The averaged RC is displayed in Fig. 1c. The same wavelength range was used to compute the average of the absolute value of MCD. The averaged MCD is displayed in Fig. 2a. Details of the analysis are illustrated in Supplementary Fig. 4. The magnetic susceptibility was evaluated from the slope of MCD at small magnetic fields ($|B| \leq 0.5$ T). The magnetic saturation field $B_s$ was defined as the field, at which the MCD reaches 85% of the saturation value. To evaluate the differential susceptibility $\frac{dMCD}{dB}$ (Fig. 3b), we applied a Savitzky–Golay filter to the experimental data before taking the numerical derivative with respect to the magnetic field.

### Determination of the phase boundaries

The boundary between the layer-polarized and layer-hybridized regions in the $(\nu, E)$ phase space was identified by the strongest dependence of the optical reflection of the moiré exciton on the electric field[25]. For $0.4 \leq \nu \leq 1$, the boundary is sharp, and the electric-field derivative of the reflection displays a clear peak. An example is shown in Supplementary Fig. 3 for $\nu = 1$. The electric-field derivative of the moiré exciton reflection shows a pronounced peak at $E_c = 60$ meV/nm. The dashed green line in Fig. 1c is a cubic spline

extrapolation of the boundary electric field to the origin. The dashed black line in Fig. 2c provides a guide to the eye of the boundary of the region with enhanced magnetic susceptibility. We define the boundary at which MCD (at $B = 0.5$ T) drops to 0.03.

### Exciton sensing

Monolayer $WS_2$ was used as the sensor. Changes in the sample compressibility modulate the dielectric environment for the sensor[39], which was probed by its 2 s exciton response as demonstrated in Refs. 35,45–47. The alignment of the sensor and sample valence bands is such that the sensor remains charge neutral for the entire range of $(\nu, E)$ throughout this study. An example is illustrated in Supplementary Fig. 4c for $E = 0$. The RC spectrum of the sensor around the 2 s resonance is displayed as a function of filling factor of the sample (the reference spectrum was acquired when the sensor is electron-doped). Both the 2 s resonance energy and intensity vary strongly with filling factor of the sample. We used the 2 s resonance intensity to represent the sample incompressibility (Fig. 1d). To extract the 2 s resonance intensity, we first removed a broad third-order polynomial background for the entire spectral window where both 2 s and higher lying excitonic resonances of the sensor are present. We then integrated around the 2 s exciton peak (black dots, Supplementary Fig. 4d) over a 2-nm wavelength window for each filling factor.

### Twist angle calibration

We calibrated the moiré density $n_M$ and the twist angle $\theta$ of $WSe_2$ bilayers using the quantum oscillations observed optically under an out-of-plane magnetic field of 8.8 T. The details for the sample examined in the main text are shown in Supplementary Fig. 2. Supplementary Fig. 2a is the MCD spectrum of the sample near the moiré exciton resonance as a function of gate voltage (or equivalently, hole density). The MCD signal oscillates due to the formation of the Landau levels (LLs). The LL period is determined to be 0.3 V (Supplementary Fig. 2b), from which we deduce a hole density change of $7 \times 10^{11}$ cm$^{-2}$ per volt. In addition, we identified insulating states through the sensor response as a function of gate voltage and assign the first four most prominent ones to be $\nu = 1/4$, 1/3, 1, and 2. Supplementary Fig. 2c shows these insulating states in filling factor and gate voltage. From the data below filling 1, we determined $n_M = (2.4 \pm 0.1) \times 10^{12}$ cm$^{-2}$ and $\theta = (2.7 \pm 0.1)$ degrees. The results also allow us to determine the hBN thickness for the top and bottom gates: $d_{tg} \approx 18$ nm and $d_{bg} \approx 15$ nm. Using these values we determined the out-of-plane electric field, $E = V_{tg}/2d_{tg} - V_{bg}/2d_{bg} - E_0$, where $V_{tg}$ and $V_{bg}$ denote the top and bottom gate voltages, respectively, and $E_0 \approx 13$ mV/nm is a built-in field likely from the asymmetry of the device structure in the presence of the sensor layer. We also note that the LL period in gate voltage decreases slightly with increasing filling factor. The nonlinear gating effect likely arises from the non-ohmic contact. We accounted for this effect by using a two-piece linear interpolation of the experimental data points above and below filling factor 1 (Supplementary Fig. 2c).

### Energy scale for magnetic saturation at $\nu = 1$

Supplementary Fig. 9 shows the temperature ($T$) dependence of the inverse magnetic susceptibility ($1/\chi$) measured by the small-field MCD for $\nu = 1$. The dependence is well described by the Curie-Weiss law ($\frac{1}{\chi} \sim T - T_{CW}$) above 10 K with a Curie-Weiss temperature of $T_{CW} \approx -(4.0 \pm 0.6)$ K. The negative sign indicates an antiferromagnetic interaction for the localized magnetic moments of the Mott insulator. The strength of the exchange interaction is estimated by $J \approx -k_B T_{CW} \approx 0.34$ meV, where $k_B$ is the Boltzmann constant. The Zeeman energy ($g\mu_B B$) from the externally applied magnetic field must overcome $J$ to achieve magnetic saturation. Here $g \approx 10$ is the hole g-factor for $WSe_2$ and $\mu_B$ is the Bohr magneton. The magnetic saturation is thus expected when $g\mu_B B \gtrsim J$ or $B \gtrsim \frac{J}{g\mu_B} \approx 0.6$ T. The field value is

consistent with the observed saturation field of 1.5 T at $\nu = 1$ from the MCD measurement (Fig. 3b).

## Band structure calculations

We used the continuum model for twisted TMD homobilayers following Refs. 8,9 to compute the single-particle band structure and the DOS of tWSe$_2$. Specifically, the moiré Hamiltonian for the valence band states at the K-valley reads

$$H_K = \begin{pmatrix} \frac{-\hbar \mathbf{k}^2}{2m^*} + \Delta_b(\mathbf{r}) + \frac{V_z}{2} & \Delta_T(\mathbf{r}) \\ \Delta_T^\dagger(\mathbf{r}) & \frac{-\hbar \mathbf{k}^2}{2m^*} + \Delta_t(\mathbf{r}) - \frac{V_z}{2} \end{pmatrix}. \tag{1}$$

The moiré Hamiltonian for the K'-valley states is a time-reversal copy of $H_K$. Here $\mathbf{k}$ is the momentum, $m^* = 0.43\, m_0$ is the effective hole mass of WSe$_2$ ($m_0$ denoting the free electron mass), $\Delta_{b,t}(\mathbf{r})$ is the bottom (top) layer energy and $\Delta_T(\mathbf{r})$ is the interlayer tunneling amplitude as a function of the spatial position $\mathbf{r}$ in the moiré unit cell. An interlayer potential difference $V_z$, which can be tuned by the out-of-plane electric field, is also introduced. In the long moiré period limit, $\Delta_{b,t}(\mathbf{r})$ and $\Delta_T(\mathbf{r})$ are smooth functions of $\mathbf{r}$ and can be approximated as $\Delta_{b,t}(\mathbf{r}) = 2V \sum_{j=1,3,5} \cos(\mathbf{G}_j \mathbf{r} \pm \psi)$ and $\Delta_T(\mathbf{r}) = w\left(1 + e^{i\mathbf{G}_2 \mathbf{r}} + e^{i\mathbf{G}_3 \mathbf{r}}\right)$ which satisfy all the symmetry constraints of the moiré superlattice. Here $\mathbf{G}_j$ is the reciprocal lattice vectors (lattice constant $a = 3.317\,\text{Å}$) and $(V, \psi, w) = (9\,\text{meV}, 128°, 18\,\text{meV})$ from Ref. 9. describe the moiré depth, shape and interlayer tunneling strength, respectively. The Hamiltonian was cut off at the 5th shell in momentum space. The computed DOS was smoothed using a Gaussian filter with a full width at half maximum of 1 meV. To compare to experiments, we converted the interlayer potential difference to the electric field using a dipole moment of 0.26 e·nm[48]. To include the effect of an out-of-plane magnetic field, we added a Zeeman energy shift between the K- and K'-valleys, using a hole $g$-factor of 10. Specifically, the band structure was first calculated without accounting for the magnetic field; the bands for the K- and K'-valley states were then displaced in energy by 0.58 meV per tesla and combined to obtain an approximation to the band structure at finite magnetic fields.

## Data availability

The Source Data underlying the figures of this study are available with the paper. All raw data generated during the current study are available from the corresponding authors upon request. Source data are provided with this paper.

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

## Acknowledgements

We thank Kaifei Kang, Liguo Ma and Liang Fu for fruitful discussions. This work was supported by the US National Science Foundation under DMR-2114535 (sensor design) and DMR-1807810 (magneto optical measurements). It was also funded partially by the US Air Force Office of Scientific Research under award no. FA9550-19-1-0390 (sample fabrication) and FA9550-20-1-0219 (modeling). The growth of hBN crystals was supported by the Elemental Strategy Initiative of MEXT, Japan, and CREST (JPMJCR15F3), JST. This work used the Cornell NanoScale Facility supported by NSF grant NNCI-2025233. We also acknowledge support from the David and Lucille Packard Fellowship (K.F.M.) and the Swiss Science Foundation Postdoc Fellowship (P.K.).

## Author contributions

P.K. and J.Z. fabricated the devices, performed the measurements, and analyzed the data. Z.X. and Y.Z. provided data from additional devices. Y.X. and Z.H. provided electronic transport measurements. K.W. and T.T. grew the bulk hBN crystals. K.F.M. and J.S. oversaw the project. All authors discussed the results and commented on the manuscript.

## Competing interests

The authors declare no competing interests.
