## [Transparent Peer Review file · Nature Communications]

Correlated states controlled by a tunable van Hove singularity in moiré WSe₂ bilayers

Corresponding Author: Professor Kin Fai Mak

Version 0:

Reviewer comments:

Reviewer #2

(Remarks to the Author)

This paper illustrates two electronic characteristics of twisted WSe₂ that track the van Hove singularity in the Moire band as it is tuned by electric field E:

First, the sample is ferromagnetic when the Fermi energy is at the E=0 vHs, and has enhanced susceptibility (though not ferromagnetic) when E_F is at the finite-E vHs.

Second, the correlated insulating state that appears at $\nu=1$ due to interactions switches from Chern C=0 at low field, where the sample is not spin-valley polarized, to co-existing C=0 and C=1 when the sample becomes spin-valley polarized by the magnetic field. The effect of the magnetic field is two-fold, inducing polarization purely via the Zeeman effect and also by tuning the vHs to cross the Fermi energy at $\nu=1$.

The data is clear, and should be published. Whether that is in Nat Comm or elsewhere is not clear to me. One thing the authors could improve would be to highlight what aspects of the discoveries are new in their paper and, more importantly, what fundamental questions they answer.

A few details that could improve the paper:

- In Fig 1b it would be helpful to show the band in 2D with contours, to make the position of the saddle point and therefore the vHs more clear
- The authors could more clearly distinguish the importance of the vHs itself vs the high DOS that goes along with it in inducing the various effects that they see. The DOS calculations they show indicate broad features, whereas some of the transitions in the data are much sharper. Is this because of a threshold effect or because of a change in the character of the Fermi surface at the vHs?
- The readability of this paper could be significantly improved with more careful attention to the writing
- Is (in)compressibility the precise quantity that is probed by exciton sensing? If not, please be more precise when describing that technique.
- The authors write, describing Fig 1c "At a critical field, there is an abrupt change in the reflection signal integrated over the intralayer exciton resonances." I do not see any abrupt change or critical field in the data. This part of the text should be rewritten to make the language more consistent with the data. The transitions drawn by dashed lines in Fig 1c look to me like guides to the eye.
- I would appreciate a discussion of whether it is expected that the spin-valley polarized state will remain for $B \gg B_c$, that is, for B large enough that the field-tuned vHs is no longer at $\nu=1$. Is the system polarized due to the Zeeman energy in the field at that point? A consideration of energy scales would be useful.

Reply to Reviewers' Comments

Reviewer #2 (Remarks to the Author):

This paper illustrates two electronic characteristics of twisted WSe₂ that track the van Hove singularity in the Moire band as it is tuned by electric field E:

First, the sample is ferromagnetic when the Fermi energy is at the E=0 vHs, and has enhanced susceptibility (though not ferromagnetic) when E_F is at the finite-E vHs.

Second, the correlated insulating state that appears at $\nu=1$ due to interactions switches from Chern C=0 at low field, where the sample is not spin-valley polarized, to co-existing C=0 and C=1 when the sample becomes spin-valley polarized by the magnetic field. The effect of the magnetic field is two-fold, inducing polarization purely via the Zeeman effect and also by tuning the vHs to cross the Fermi energy at $\nu=1$.

The data is clear, and should be published. Whether that is in Nat Comm or elsewhere is not clear to me. One thing the authors could improve would be to highlight what aspects of the discoveries are new in their paper and, more importantly, what fundamental questions they answer.

Response 1:

We thank the reviewer for the comments. Below we summarize the new findings in this study.

1. We reported a magnetic metal phase in twisted bilayer WSe₂ for the first time by exploring a twist angle range that has not been studied before. This adds a new entry to our materials library for building devices relying on ferromagnetism or topological edge states.
2. We elucidated the underlying mechanism for the emergent magnetism by carefully studying the evolution of the band structure vHS with doping and electric field. This is achieved through combined magneto-optical and electronic compressibility measurements and continuum model band structure calculations.
3. We demonstrated the importance of the vHS in determining the nature of the correlated insulating state at $\nu = 1$. The state transitions from a valley-coherent Mott insulator to competing spin-polarized Mott insulator and Chern insulator when the vHS is tuned across $\nu = 1$ by electric and magnetic fields.
4. We examined the twist angle dependence over an angle range with intermediate correlation strength, where Stoner ferromagnetism emerges. This contrasts with earlier studies, which have focused on either the small twist angle range with very strong correlations (Ref. 8) or the large twist angle range with weaker correlation effects that do not support magnetism (Ref. 1 and 2).

We note that these results are particularly relevant to the recent discovery of superconductivity and the Mott insulator-to-superconductor quantum phase transition in twisted bilayer WSe₂ (Ref. 1 and 2) because the nature of the correlated insulator at $\nu = 1$ and its interplay with the location of the vHS are expected to play an important role in understanding the emergent superconductivity. The proximity of the ferromagnetic ground state to the superconducting state in the phase diagram may also have important implications. We therefore believe that our results will be of interest to

the general readers of Nature Communications. We have improved the introduction and conclusion in the revised manuscript to further bring out the points mentioned above.

A few details that could improve the paper:

- In Fig 1b it would be helpful to show the band in 2D with contours, to make the position of the saddle point and therefore the vHs more clear

Response 2:

We thank the reviewer for the suggestion. Instead of a 2D contour plot, we now include the band dispersions along both the $\kappa - m$ and the $\gamma - m$ directions so that the saddle point singularity at the m point of the moiré Brillouin zone is clearly shown.

- The authors could more clearly distinguish the importance of the vHs itself vs the high DOS that goes along with it in inducing the various effects that they see. The DOS calculations they show indicate broad features, whereas some of the transitions in the data are much sharper. Is this because of a threshold effect or because of a change in the character of the Fermi surface at the vHs?

Response 3:

We thank the reviewer for the suggestion. The reviewer is correct that the observed ferromagnetism is of a threshold effect. Spontaneous symmetry breaking is expected to occur only when the DOS (D_F) exceeds a critical value. The relevant criterion in this case is the Stoner criterion, $D_F U > 1$, where the dimensionless interaction strength $D_F U$ is given by the product of the DOS and the on-site Coulomb repulsion U . The dimensionless interaction strength needs to exceed unity to stabilize a ferromagnetic ground state. Therefore, even the high DOE feature appears broad, the ferromagnetic transition can be sharp. On the other hand, even when the DOS is not large enough to stabilize a ferromagnetic ground state, the vHS (where the DOS is maximized) is still able to significantly enhance the magnetic susceptibility of the system, as observed in Fig. 2c. In the revised manuscript, we have clearly mentioned the two effects in the last paragraph of the section on “Stoner ferromagnetism”, namely, a maximum DOS at the vHS that significantly enhanced the magnetic susceptibility and a critical DOS that satisfies the Stoner criterion and stabilizes a ferromagnetic ground state.

- The readability of this paper could be significantly improved with more careful attention to the writing

Response 4:

We thank the reviewer for the suggestion. We have carefully modified our manuscript to make it more accessible to the readers.

- Is (in)compressibility the precise quantity that is probed by exciton sensing? If not, please be more precise when describing that technique.

Response 5:

The exciton sensing technique provides a proxy for the electronic incompressibility through measurements of the dielectric constant of the sample, which screens the exciton binding energy in the sensor layer (see Nature Communications 13, 4271 (2022) and Nature 587, 214-218 (2020) for details of the exciton sensing technique). It is not a direct measurement of the electronic incompressibility. We have described the technique more precisely in the revised manuscript.

- The authors write, describing Fig 1c "At a critical field, there is an abrupt change in the reflection signal integrated over the intralayer exciton resonances." I do not see any abrupt change or critical field in the data. This part of the text should be rewritten to make the language more consistent with the data. The transitions drawn by dashed lines in Fig 1c look to me like guides to the eye.

Response 6:

We thank the reviewer for the comment. The boundary between layer-hybridized and layer-polarized regions is determined by the maximum and minimum in the electric-field derivative of the reflection contrast, as shown in Extended Data Fig. 3. Indeed, it is not an abrupt change in the reflection contrast; we have used the maximum in the electric-field derivative to determine the boundary. In the revised manuscript, we have modified the text in the third paragraph of the section on "Phase diagram of tWSe₂" to precisely describe the criterion used to determine the phase boundary.

- I would appreciate a discussion of whether it is expected that the spin-valley polarized state will remain for $B \gg B_c$, that is, for B large enough that the field-tuned vHs is no longer at $\nu=1$. Is the system polarized due to the Zeeman energy in the field at that point? A consideration of energy scales would be useful.

Response 7:

We thank the reviewer for the question. The system is spin-valley-polarized at high magnetic fields near filling factor 1, as shown by the saturated MCD response in Fig. 3b. To compare the relevant energy scales at filling 1, we show in Fig. R1 the temperature dependence of the inverse magnetic susceptibility measured by the small-field MCD. The dependence is well described by the Curie-Weiss law with a Curie-Weiss temperature of $T_{CW} \approx -4$ K. The negative sign shows an antiferromagnetic interaction for the localized magnetic moments of the Mott insulator. The strength of the exchange interaction is given by $J \approx -k_B T_{CW} \approx 0.34$ meV (k_B is the Boltzmann constant). The Zeeman energy ($g\mu_B B$) due to the external field has to overcome this energy scale to achieve a fully polarized state. (Here $g \approx 10$ is the hole g -factor in WSe₂ and μ_B is the Bohr magneton.) Full polarization is expected when $g\mu_B B \gtrsim J$ or $B \gtrsim \frac{J}{g\mu_B} \approx 0.6$ T, which is consistent with the observed saturated MCD for $B \gtrsim 1.5$ T at filling 1 (Fig. 3b). We have included this estimate in the revised Methods and Fig. R1 as Supplementary Fig. 9.

Figure R1. Temperature dependence of the inverse magnetic susceptibility. The experimental data (symbols) are well described by the Curie-Weiss law (solid lines) above 10 K with a Curie-Weiss temperature of $-(4.0 \pm 0.6)$ K for $\nu = 1$ and 4.0 ± 0.2 for $\nu = 0.8$.